# Molecular Identification of Trypanosome Diversity in Domestic Animals Reveals the Presence of *Trypanosoma brucei gambiense* in Historical Foci of Human African Trypanosomiasis in Gabon

**DOI:** 10.3390/pathogens11090992

**Published:** 2022-08-30

**Authors:** Larson Boundenga, Illich Manfred Mombo, Mouinga-Ondeme Augustin, Ngoubangoye Barthélémy, Patrice Makouloutou Nzassi, Nancy D. Moukodoum, Virginie Rougeron, Franck Prugnolle

**Affiliations:** 1International Centre for Medical Research in Franceville (CIRMF), Franceville BP 769, Gabon; 2Department of Anthropology, Durham University, South Road, Durham DH1 3LE, UK; 3Department of Animal Biology and Ecology, Tropical Ecology Research Institute (IRET-CENAREST), Libreville BP 13354, Gabon; 4REHABS, International Research Laboratory, CNRS-NMU-UCBL, George Campus, Nelson Mandela University, George 6529, South Africa

**Keywords:** trypanosomes, diversity, prevalence, domestic animals, *Trypanosoma brucei gambiense*, Gabon

## Abstract

Human African Trypanosomiasis (HAT) is an infectious disease caused by protozoan parasites belonging to the *Trypanosoma* genus. In sub-Saharan Africa, there is a significant threat as many people are at risk of infection. Despite this, HAT is classified as a neglected tropical disease. Over the last few years, several studies have reported the existence of a wide diversity of trypanosome species circulating in African animals. Thus, domestic and wild animals could be reservoirs of potentially dangerous trypanosomes for human populations. However, very little is known about the role of domestic animals in maintaining the transmission cycle of human trypanosomes in central Africa, especially in Gabon, where serious cases of infection are recorded each year, sometimes leading to hospitalization or death of patients. Komo-Mondah, located within Estuaries (Gabonese province), stays the most active HAT disease focus in Gabon, with a mean of 20 cases per year. In this study, we evaluated the diversity and prevalence of trypanosomes circulating in domestic animals using the Polymerase Chain Reaction (PCR) technique. We found that 19.34% (53/274) of the domestic animals we studied were infected with trypanosomes. The infection rates varied among taxa, with 23.21% (13/56) of dogs, 16.10% (19/118) of goats, and 21.00% (21/100) of sheep infected. In addition, we have observed a global mixed rate of infections of 20.75% (11/53) among infected individuals. Molecular analyses revealed that at least six *Trypanosome* species circulate in domestic animals in Gabon (*T. congolense*, *T. simiae*, *T. simiae* Tsavo, *T. theileri*, *T. vivax*, *T. brucei* (including *T. brucei brucei*, and *T. brucei gambiense*)). In conclusion, our study showed that domestic animals constitute important potential reservoirs for trypanosome parasites, including *T. brucei gambiense*, which is responsible for HAT.

## 1. Introduction

Human African trypanosomiasis (HAT) or sleeping sickness is an infectious disease caused by protozoan parasites belonging to the genus *Trypanosoma* [1]. They are transmitted to humans by the bite of a hematophagous fly of the genus *Glossina* (tsetse fly) [2]. In sub-Saharan Africa, there is a significant threat as many people are at risk of infection. In this region, transmissions are largely observed in rural populations and among people engaged in activities like agriculture, hunting, and fishing [3,4]. During the late 20th century, approximately 300,000 to 500,000 people were estimated to have died from HAT [5,6]. Moreover, the number of people at risk, i.e., potentially exposed to this disease, is estimated at 65 million [7]. In Africa, there are three subspecies of *Trypanosoma brucei*: *T. b. brucei*, *T. b. gambiense,* and *T. b. rhodesiense*, of which two infect humans (*T. b. gambiense* and *T. b. rhodesiense)* [8,9]. The disease caused by these agents is often fatal if untreated [10]. 

Thus, to control HAT, numerous initiatives and programs were developed in the African continent in the mid-1960s [11,12]. Strategies of control for this vector-borne disease are essentially based on active disease surveillance through periodic screening programs, followed by treatment of found patients, supported by tsetse fly vector control [11,13]. The fight against HAT appeared to be making considerable progress. However, a major resurgence of the disease has been observed since the late 1990s [11]. This is believed to be the result of profound environmental changes, a considerable decline in medical surveillance of historical outbreaks in many African countries, and the movement of livestock to new habitats [11]. Today, it is clear that when vector (tsetse fly) control is simultaneously conducted with human and animal reservoir identification strategies, it results in an accelerated elimination of the disease [12,14].

African Animal Trypanosomiasis (AAT) is also called Nagana and affects both wild and domestic animals [15,16,17]. Several studies have reported the existence of a wide diversity of trypanosome species circulating in African animals: *T. brucei*, *T. congolense*, *T. vivax*, *T. simiae*, *T. godfrey**i*, and *T. theil**eri* [17,18,19,20]. Recent studies have reported serious human infections with trypanosome parasites, such as *T. congolense*, *T. vivax*, *T. b. brucei,* and *T. lewisi*, which naturally infect animals [21,22,23]. Thus, animals act as reservoirs of potentially dangerous trypanosomes for human populations [24,25]. However, very little is known about the role of domestic animals in maintaining the cycle or transmission of human trypanosomes of *T. b. gambiense* in humans [26,27].

Very little information is available concerning the diversity of trypanosome species circulating in wildlife or domestic fauna in Gabon [18,19], although several ancient foci are known to still be active [28,29]. HAT was reported to be present in seven out of the nine provinces of this country during the 20th century [30]. Serious cases of infection are recorded each year, sometimes leading to hospitalization or death of patients [29]. Komo-Mondah, located within the province of Estuaire, is the most active focus of HAT, with a mean of 20 cases per year [29,31]. Estuaire province is infested by tsetse flies [31], and the expansion of cities has brought urban populations closer to tsetse fly sites, which may expose populations to HAT due to more regular contact between humans and tsetse flies. In addition, the expansion of cities marked by the encroachment and modification of forest ecosystems creates new environments where humans and animals are exposed to the same disease vectors. This situation could contribute to promoting the transmission of numerous zoonotic pathogens as well as favor a high exposure of human populations to animal parasites (animal trypanosomes in our case) that could have consequences for public health [32,33]. Thus, it is important to determine the diversity of trypanosome species and their levels of infection among animals sharing the same environments with human populations to improve protocols for controlling this disease. In this study, we evaluated the diversity of trypanosomes circulating in domestic animals in four provinces of Gabon to assess the risks to the human population.

## 2. Results

### 2.1. Occurrence of Trypanosomes

In 274 blood samples from domestic animals, the overall frequency of trypanosome infection was 19.34% (53/274). Among the 53 individuals positive with *Trypanosoma* spp., 23.21% (13/56) were dogs, 16.10% (19/118) were goats, and 21% (21/100) were sheep (Table 1). However, the difference in infection rate between dogs and small ruminants (17% overall) was not statistically significant (*p* = 0.42). Infection rates and trypanosomes species also varied from site to site (Table 1). 

### 2.2. Diversity and Prevalence of Each Parasite

Of 53 individuals infected with trypanosome species, we found that 2% (7/274) were infected with *T. b. gambiense* subspecies (Table 2). Molecular analyses revealed that six *Trpyanosoma* species circulate in these domestic animals. We detected two trypanosome species in dogs (*T. congolense and T. brucei,* in particular *T. b. brucei* and *T. b. gambiense* subspecies), four in sheep (*T. congolense, T. simiae, T. theileri* and *T. brucei* in particular *T. b. gambiense subspecies*), and five in goats (*T. congolense, T. simiae, T. vivax, T. simiae* Tsavo and *T. brucei* in particular *T. b. brucei*) (Table 2). The prevalence of each parasite species varied with host species (Table 2). *T. b. gambiense* was only observed in dogs and sheep. However, dogs were more infected than sheep, with 9% (Table 2). The distribution of trypanosomes species was almost identical in all cities we studied. However, *T. b. gambiense* subspecies were found only in the old foci of this disease: Estuaire, Ogooue-Maritime, and Moyen-Ogooue (Figure 1).

### 2.3. Prevalence of the Mixed Infections

The global mixed infections rate was 21% (11/53). Goats exhibited the highest mixed infection rate of 36.84% (7/19), followed by dogs at 15.38% (2/13), and sheep with 9.52% (2/21) (Table 3). These mixed infections included various combinations of parasite. In goats, the most predominant association was *T. simiae*-*T. simiae* Tsavo (*n* = 3), while no association was predominant in dogs and sheep (Table 3).

### 2.4. Overview of Some African Countries Reported Isolation of T. b. gambiense in Domestic Animals

The census of countries that reported the presence of *T. b. gambiense* in domestic animals allowed us to produce a spatial distribution map of different trypanosome species identified in animals and *T. b. gambiense* subspecies too (Figure 2). This spatial visualization shed light on the distribution of studies that report the presence of trypanosomes causing diseases in humans beyond the Central African sub-region and different host species (sheep, goats, dogs, cats, and cattle) living in contact with the carrier populations of these parasites.

## 3. Discussion

Animal trypanosomiasis occurs throughout the tropical regions of Africa. This disease affects many mammal species, including domestic animals like dogs, sheep, goats, pigs and cattle [18,19,20,60]. The role of these animals in the maintenance and transmission of the disease in humans is not well known [24,43]. The lack of data concerning Central Africa, in particular Gabon, and the need to put appropriate strategies in place to combat this disease are both reasons for studying trypanosomiasis in domestic animals. Our study estimates the prevalence of trypanosome infection in domestic animals and attempts to identify the diversity of species circulating in four provinces of Gabon.

### 3.1. Diversity of Trypanosomes

Molecular and phylogenetic analyses revealed that different groups of domestic animals (goats, sheep, and dogs) are infected with several *Trypanosoma* species. We found six trypanosome species in domestic animals (*T. simiae*, *T. simiae* Tsavo, *T. vivax, T. congolense*, *T. theilerie*, *and T. brucei* (mainly *T. b. brucei*, and *T. b. gambiense*)). All species identified in our study were previously found in domestic animals [39,77,78,79]. These similar results could be explained by the fact that the vectors of these parasites are widely spread throughout the African continent and by the movement of animals from one country to another favored by breeding and trading [80]. We also observed that goats and dogs entered the forest regularly for various activities (looking for food, fleeing from danger, hunting and agricultural activities) during sampling, sometimes with their owners. These activities may explain the diversity of parasites found in these domestic animals.

Species richness was higher in goats, with five species identified (*T. vivax, T. simiae, T. simiae* Tsavo, *T. congolense* and *T. brucei (T. b. brucei* subspecies)), four species in sheep (*T. simiae, T. theileri, T. congolense* and *T. brucei* (*T. b. gambiense* subspecies)), and two species in dogs (*T. congolense*
*and T. brucei* whose *T. b. brucei* and *T. b. gambiense* subspecies). Infection rate was low (<10%) for parasite species in different host groups, which is consistent with the literature [43]. In goats, the predominant species were *T. vivax* at ~6% and *T. simiae* with 4%. In sheep, the predominant species were *T. simiae* with 8% and *T. theilerie* with 7%. In dogs *T. brucei* species in particular *T. b. brucei*
*and T. b. gambiense* subspecies were predominant with about 9% infection rate. Of these, three predominant species (*T. brucei*, *T. vivax,* and *T. simiae*) are known to have deleterious effects on the health of pets [81]. Thus, it is important to treat animals to reduce the impact of these parasites on their health and to potentially break the chain of parasite transmission. Some of the parasites we identified are common in other animals. Thus, our study supports the hypothesis that domestic animals are a reservoir of a large diversity of trypanosomes in both animals [39,43] and humans [21,23] because cases of infections of humans with animal trypanosomes species can occur [23].

In our study, *T. b. gambiense* was found in dogs and sheep with infection rates of 9% and 2%, respectively. These observations support the existence of a potential domestic animal reservoir for *T. b. gambiense* HAT in Central Africa [24,39,43]. To our knowledge, this study is the first to report the presence of *T. b. gambiense* in animals from HAT outbreaks in Gabon, particularly in dogs and sheep. We only observed animals infected with *T. b. gambiense* in areas known to be historical foci of HAT. The existence of an animal reservoir of trypanosomes in Gabon could explain the failure of elimination strategies for trypanosomiasis and may be responsible for the sporadic cases that frequently re-emerge in most of the historical foci [29,60,72]. Thus, the presence of *T. b. gambiense* in dogs and sheep could be explained by the fact that they live in ancient foci of HAT in close proximity to their owners and their families, favoring parasite exchange, or by the fact that they are nomadic animals and may frequent environments where they are exposed to the parasite vectors [19,43]. In this context, dogs and sheep are more likely to be involved in the transmission cycle involving human populations in the different foci found in these four provinces, and thus are themselves representing potential vectors of *T. b. gambiense*. Any strategy to control trypanosomiasis should consider animals living close to human populations. The systematic screening of domestic animals in these places is very important to determine the prevalence of this disease. Moreover, since these animals can serve as a bridge between natural environments (forests and savannas) and the anthropized environment (villages or cities), it is also important to expand research on wild animals. This would make it possible, on the one hand, to describe the diversity of trypanosomes circulating in nature and, on the other hand, to identify potential host species or reservoirs. Undertaking studies on tsetse fly bloodmeals from different HAT outbreaks in Gabon would be interesting to better understand the frequency of contact between tsetse flies and domestic animals. The epidemiological implications of these animals may vary depending on the epidemiological context of the different HAT foci. 

### 3.2. Prevalence of Trypanosoma spp. Infection

We showed that domestic animals have a high rate of trypanosome infection, with an overall infection rate of 19.34% (53/274). This is consistent with previous studies in Tanzania, Ghana, and Kenya where the prevalence was 16.7%, 17.4%, and 21%, respectively [25,52,82]. However, the infection rate in our study is low compared to that obtained in domestic animals in Nigeria (46.8%) and Cameroon (35.2%) [83,84]. Differences observed between our results, and some previous studies may be related to ecological factors intrinsic to each region or to geographical distribution. 

The level of infection varies from one species to another and from one site to another. Concerning host species, infection rates were higher in sheep (21.0%) and dogs (23.21%) than in goats (16.10%). In contrast, dogs (23.21%) appeared to have a higher level of infection than small ruminants (goats and sheep), which had a rate of 18.34%. Nevertheless, this difference in infection rates between dogs and small ruminants was not significant. The latest could be explained by the fact that dogs are more exposed to tsetse bites through various activities, such as hunting and agriculture during which they accompany humans [43]. The low infection rates in small ruminants may be partly attributed to the fact that some small ruminants, like goats, show defensive behavior against the bites of tsetse flies. For instance, they could produce less odor, which might reduce the number of tsetse flies [43,85,86,87]. In addition, the length of hair on different animals, such as sheep, could limit the bite of tsetse flies. Indeed, if the sheep are quite hairy, the tsetse flies will have more difficulty having a blood meal and thus transmitting the trypanosomes.

The three cities (Ntoum (22%), Cocobeach (24%), Port-Gentil (27%)), which had an overall infection level of more than 20%, are historical foci of HAT in Gabon. The observation of animals infected by trypanosomes shows that these outbreaks are still active and justifies the need for surveillance measures [28,29]. In addition, the rate of 18% and 20%, respectively, observed in Libreville and Franceville, supports the idea of active transmission of HAT in urban areas in Gabon, as previously observed in the human population [28].

Although the level of infection per species is low, *T. simiae*, *T. b. brucei,* and *T. congolense* had the highest infection rate among domestic animals with 4.74%, 3.28%, and 3.28%, respectively. This observation does not fully support previous studies, suggesting that the species most frequently encountered in domestic animals of central Africa, and other African areas are *T. congolense* and *T. vivax* [84,88,89,90]. These predominant species could have a strong impact on livestock farmers as they cause enormous damage to domestic animals. *T. simiae* and *T. congolense* are considered highly pathogenic for livestock and can cause high mortality in some animals [91,92]. In addition, the presence of these parasites (*T. congolense* and *T. b. brucei*) may constitute a risk to the health of human populations, as they can infect humans [23,93].

We found some individuals with mixed infections of trypanosome parasites, which are very dangerous to domestic animals. Among the mixed infections observed, the most common associations were *T. simiae*-*T. simiae* Tsavo and *T. vivax-T. congolense* in goats. For dogs and sheep, we observed no predominant association. Our results are consistent with previous studies [19,94], which showed that infected small ruminants and tsetse flies from several areas of Gabon, including the sleeping sickness foci, frequently harbor more than one trypanosome species [19,94,95]. Thus, mixed infection rates could be explained by the high exposure of animals to vectors infected by several trypanosomes species in different environments [36,52]. The observation of high levels of mixed infections in our study shows the need for further studies. Future investigations on mixed infections may enable us to understand aspects of the ecology and evolution of trypanosomes and to determine if genetic exchanges between trypanosome species can occur within hosts.

### 3.3. The Necessity of One Health Approach Strategies

Finally, from an ecological point of view, we could say that, unlike its counterpart *T. b. rhodesiense* for which the zoonotic potential has been established [49], this has yet to be determined for *T. b. gambiense* and not clearly defined [7,49]. However, this parasite can certainly infect several species of domestic animals, including dogs, goats, sheep, pigs, cats, and cattle, which, in turn, can serve as infectious reservoirs for humans [7]. Although domestic animals may not have a significant role in the transmission of *T. b. gambiense* to humans, the existence of a potential domestic animal reservoir could have implications on the control of its transmission to humans. Thus, we believe that animals harboring the parasite and sharing the same environment with human populations may play an important role in the epidemiology of trypanosomiasis or in maintaining interpopulation transmission.

The dataset from research reporting the presence of *T. b. gambiense* in domestic animals in sub-Saharan Africa (Figure 2) highlighted that this observation is generalized in some historical foci of the HAT and suggests that it is important to better understand the role of domestic animals in the epidemiology of HAT in West and Central African regions where domestic animals are abundant, particularly in areas where these animals cohabit with humans and are also known to harbor the vectors of this parasite. Thus, any strategy to eradicate this disease should take into account the various components at the human-domestic animal-vector interface through a One Health approach.

Indeed, instead of focusing only on the main known reservoir, which is human [96], the elimination or control strategy against HAT *gambiense* should also include animals and vectors living in the same settings for effective control or elimination. However, the implementation of such a control or elimination strategy for Gambian HAT in Gabon would require clarification of the role played by domestic animals in trypanosomiasis transmission, i.e., whether the presence of these animals would decrease or increase the transmission of *T. b. gambiense* to humans as it has been done for malaria [97,98,99]. Furthermore, we believe that an optimal control strategy against HAT *gambiense* would require a better understanding of the ecology of the interactions between the different host species sharing the same environment [33,49], because adapting the control strategies to the changing patterns of disease transmission has been one of the keys to success in recent years and should continue in the future in a One Health approach [49,100].

## 4. Materials and Methods

### 4.1. Study Area and Samples Collection

To characterize trypanosome diversity in domestic animals (goat, sheep, dogs), we collected biological material from several villages near forests in four provinces of Gabon (Estuaire (Es), Ogooue-Maritime (OM), Moyen-Ogooue (MO), and Haut-Ogooue (HO)) between 2017 and 2019. Veterinarians handled the animals and took blood samples in strict compliance with animal welfare criteria. Approximately 2 to 7 mL of blood was collected in EDTA tubes according to the mass of each individual and immediately placed in a cooler and transported to the CIRMF laboratories, where it was stored at −20 °C. A total of 274 animal samples were collected, including from 118 goats, 100 sheep, and 56 dogs.

### 4.2. DNA Extraction and Polymerase Chain Reaction (PCR)

For all blood samples, total DNA for each sample was extracted using the DNeasy Blood and Tissue Kit (Qiagen, Courtaboeuf, France) from approximately 200 μL of blood according to the manufacturer’s procedures. 

### 4.3. Molecular Amplification for Identification of Different Trypanosome Species

Total DNA was then used as a template for the detection of trypanosome parasites according to previously described protocols based on the amplification of a portion of the trypanosome 18S rRNA genes using nested PCR [81]. The primers used for these protocols are described in Table 4. All amplified products (10 μL) were run on 1.5% agarose gels in Tris-acetate-EDTA buffer. Amplicons were sequenced and deposited in GenBank under the following accession numbers: (MW364063-MW364115).

### 4.4. Molecular Identification of T. b. gambiense

After identification of different trypanosomes species, we searched for *T. b. gambiense* in all samples that showed a DNA fragment of about 400 bp corresponding to the size expected for trypanosomes of the subgenus Trypanozoon. For this identification, two PCR rounds were performed. We performed the first round as described by Cordon-Obras et al. [8]. For this *T. b. gambiense* identification, we used two pairs of specific primers. The first pair (TgSGP-1: 5′–GCTGCTGTGTTCGGAGAGC–3′ and TgSGP-2: 5′–GCC ATCGTGCTTGCCGCTC–3′) was described by Radwanska et al. [101]. The first PCR round was carried out in a total volume of 25 μL containing 1× PCR buffer (Tris–10 mM HCl, 50 mM KCl, 3 mM MgCl_2_), 15 picomoles of each primer (TgSGP-1/2), 100 mM of each dNTP, one unit of Taq DNA polymerase, 5 μL of DNA extract and 14 μL of sterile water. For the first round (PCR1), the program conditions were as follows: Initial denaturation at 95 °C for 3 min. This was followed by 45 cycles made up of a denaturation step at 95 °C for 30 s, an annealing step at 63 °C for 1 min and an elongation step at 72 °C for 1 min. A final elongation was done at 72 °C for 5 min.

For the second round (PCR2), we used another primer pair (TgsGP-s: 5′–TCAGAC AGG GCT GTA ATA GCAAGC-3′ and TgsGP-as: 5′–GGGCTCCTGCCTCAATTGCTGCA–3′), which was designed by Morrison et al. [102]. For this step, we used amplicons of PCR1 as a DNA template. In this nested PCR, only 25 amplification cycles were performed in the same conditions as for the first PCR. All amplified products of PCR2 (around 10 μL for each sample) were run on 1.5% agarose gels in TAE buffer. All samples in which a DNA fragment of about around 270 bp was revealed after PCR and electrophoresis were considered positive for *T. b. gambiense*.

## 5. Conclusions

This study showed that domestic animals constitute important potential reservoirs for a large diversity of trypanosomes, including *T. b. gambiense* subspsecies of *T. brucei*, which is responsible for HAT. These animals should be considered in trypanosomiasis control strategies. Further studies are now necessary to better characterize the role of domestic animals in maintaining the transmission of this disease, assess the risk to animal and human health, and understand the ecology of these parasites in Gabon, especially the role of wild animals in their cycle.

## Figures and Tables

**Figure 1 pathogens-11-00992-f001:**
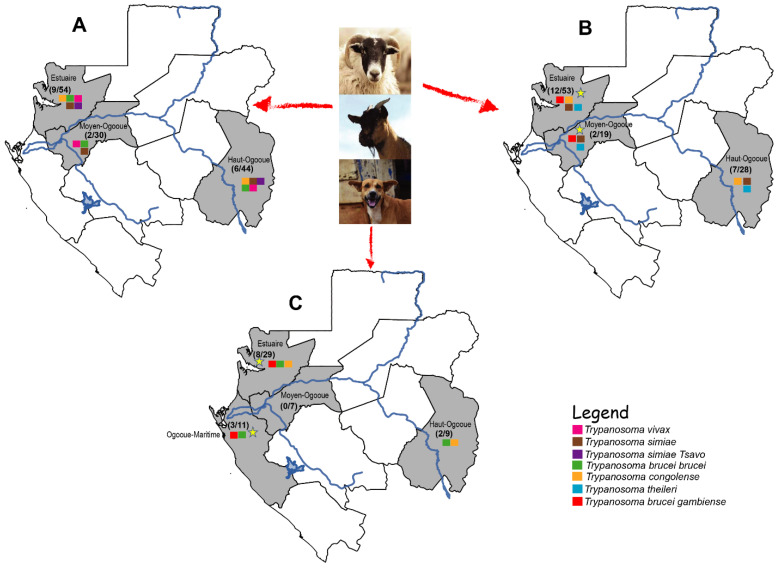
Spatial distribution of trypanosome species in different host species. (**A**) Distribution of trypanosomes identified in goats. (**B**) Distribution of parasitic trypanosomes in sheep and (**C**) the location of trypanosomes found in dogs. The yellow star indicates the provinces where *T. b. gambiense* parasite was found in domestic animals.

**Figure 2 pathogens-11-00992-f002:**
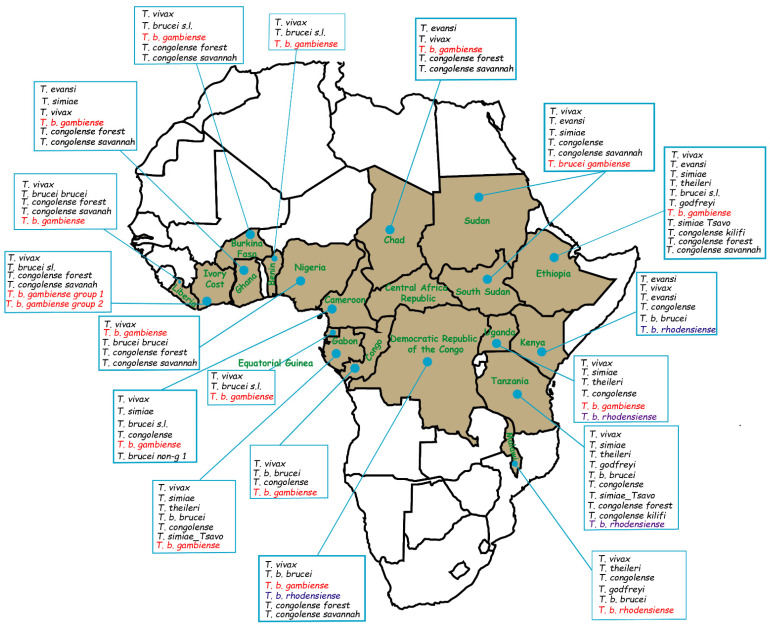
Map showing the distribution of countries reported the domestic animals infected with different trypanosome species in Central Africa sub-region and other countries. This map is based on the data provided in several studies [20,24,25,34,35,36,37,38,39,40,41,42,43,44,45,46,47,48,49,50,51,52,53,54,55,56,57,58,59,60,61,62,63,64,65,66,67,68,69,70,71,72,73,74,75,76] and the data obtained in our study. The species written in black are the species that naturally infect animals; the purple and red colors designate the subspecies of *T. brucei* that cause disease in humans in Africa.

**Table 1 pathogens-11-00992-t001:** Prevalence of trypanosomes infection by studied localities in Gabon and per hosts utilized in this study.

		Dogs		Goats		Sheep		Prevalence (%) (Infected/Examined)
Province	City	Examined	Infected (%)	Examined	Infected (%)	Examined	Infected (%)	
**Estuaire**	Libreville	14	3 (21)	12	2 (17)	19	4 (21)	20 ± 11.69 (9/45)
	Cocobeach	10	4 (40)	23	6 (26)	21	3 (14)	24 ± 11.40 (13/54)
	Ntoum	05	1 (20)	19	3 (16)	13	5 (38)	24 ± 13.82 (9/37)
**Ogooue-Maritime**	Port-Gentil	11	3 (27)	-		-		27 ± 18.53 (3/11)
**Moyen-Ogooue**	Lambarene	07	0 (0)	20	2 (10)	19	2 (11)	9 ± 8.14 (4/46)
**Haut-Ogooue**	Bongoville	04	2 (50)	10	0 (0)	17	3 (18)	16 ± 12.95 (5/31)
	Franceville	05	0 (0)	34	6 (18)	11	4 (36)	20 ± 11.09 (10/50)
	**Total prevalence**	**56**	**13 (23.21 ± 11.06)**	**118**	**19 (16.1 ± 6.63)**	**100**	**21 (21 ± 7.98)**	**19 ± 4.68 (53/274)**

**Table 2 pathogens-11-00992-t002:** Diversity and prevalence of each trypanosome species identified in animals studied.

Parasite Species/Subspecies	Dogs *n* (Pravalence)	Goats *n* (Prevalence)	Sheep *n* (Prevalence)	Global Infection Rate (%)
*T. b. brucei*	5 (9)	4 (3)	-	3.28
*T. b. gambiense*	5 (9)	-	2 (2)	2.55
*T. congolense*	3 (5)	2 (2)	4 (4)	3.28
*T. vivax*	-	8 (7)	-	2.91
*T. simiae*	-	5 (4)	8 (8)	4.74
*T. simiae* Tsavo	-	3 (3)	-	1.09
*T. theileri*	-	-	7 (7)	2.55
Total	13/56 (23.2)	22/118 (18.6)	21/100 (21)	

**Table 3 pathogens-11-00992-t003:** Relative frequency of mixed infection detected in studied domestic animals.

Host Species	Percent Mixed Infections (Co-Infected/Infected)	Parasites (Number of Observations)
Goats	36.84% (7/19)	*T. b. brucei-T. simiae (1)*
		*T. vivax-T. simiae (1)*
		*T. simae-T. simiae* Tsavo *(3)*
		*T. vivax-T. congolense (2)*
Dogs	15.38% (2/13)	*T. b. brucei-T. b. gambiense (1)*
		*T. b. brucei-T. congolense (1)*
Sheep	9.52% (2/21)	*T. simiae-T. b. gambiens (1)*
		*T. simiae-T. theileri (1)*

**Table 4 pathogens-11-00992-t004:** List of PCR primers used for diagnosis in this study.

Round	Primer Sequences	Parasites Species/Subspecies	Gene Amplified	Product Size (bp)
PCR1	ITS 1: 5′-GATTACGTCCCTGCCATTTG-3′	*T. congolense*	18S ARNr gene	1408–1501
	ITS 2: 5′-TTGTTCGCTATCGGTCTTCC- 3′	*T. brucei brucei*		1215
PCR2	ITS 3: 5′-GGAAGCAAAAGTCGTAACAAGG-3′	*T. theileri*		998
	ITS 4: 5′-TGTTTTCTTTTCCTCCGCTG-3′	*T. simiae* Tsavo		951
		*T. simiae*		847
		*T. vivax*		620
PCR1	TgSGP-1: 5′–GCTGCTGTGTTCGGAGAGC–3′	*T. b. gambiense*		270
	TgSGP-2: 5–GCC ATCGTGCTTGCCGCTC–3			
PCR1	TgsGP-s: 5′–TCAGAC AGG GCT GTA ATA GCAAGC-3′			
	TgsGP-as: 5–GGGCTCCTGCCTCAATTGCTGCA–3′			

## Data Availability

All sequences generated in this study are available in Genbank.

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
