# Peer review of "Molecular Identification of Trypanosome Diversity in Domestic Animals Reveals the Presence of Trypanosoma brucei gambiense in Historical Foci of Human African Trypanosomiasis in Gabon"

_pathogens, 2022, doi:10.3390/pathogens11090992_

Round 1

Reviewer 1 Report (Previous Reviewer 1)

The manuscript is much improved from the previous submission. Although some improvements could still be made

So for example the authors need to be clear on differences between sub species and species, so Trypanosoma vivax is a species, Trypanosoma brucei rhodesiense are sub species of T. brucei (occurs in several places in the MS)

In some of the newly added text instead of the reference by authors, they appear as PMID numbers.

Line 71, I think this section of text in  terms of reservoirs refers to T. b. gambiense rather than T. b. rhodesiense.

Line 79, infested might be better wording than infected

In the results you still  need to include 95% confidence intervals for your prevalence values.

Figure 2, are the images of the possible reservoirs examples of local breeds of sheep, goats and dogs. gambiense is spelled incorrectly in the legend

Line 137, what does ref mean, is this figure based on molecular analyses only. The wings are are probably stars

Figure 3, spelling of Equatorial Guinea and Cameroon

Line 192 places not place

Line 202 prevalence is its own plural

Line 219 check prevalence for goats and small ruminants, according to the table it should be goats 16% and small ruminants 18%. Note you have no significant difference between small ruminants and dogs so they are essentially the same. One thing you might want to note is the length of hair on the various animals as if sheep are quite hairy then this might make it more difficult for tsetse to take a meal

Line 220 there is mention of cities, how urban/rural are the study locations

Author Response

Rewiers 1

Comments and Suggestions for Authors

The manuscript is much improved from the previous submission. Although some improvements could still be made

So for example the authors need to be clear on differences between sub species and species, so Trypanosoma vivax is a species, Trypanosoma brucei rhodesiense are sub species of T. brucei (occurs in several places in the MS)

We agree with reviewer 1 in this aspect concerning T. brucei subspecies.  Now T. b. gambiense or T. b. rodhesiens were edited in all main text of new manuscript version. Indeed, we replaced speceis by subspecies in all part of manuscript which contain this subspeceis of T. brucei.

In some of the newly added text instead of the reference by authors, they appear as PMID numbers.

Thank you, we agree and we edited in all manuscript version (see line 86 ; 139)

Line 71, I think this section of text in terms of reservoirs refers to T. b. gambiense rather than T. b. rhodesiense.

We agree with reviewer 1 and we edited this sentence in new manuscript (see line 71)

Line 79, infested might be better wording than infected

Thank you, now we replaced infected by infested (see line 79 of new manucript version)

In the results you still need to include 95% confidence intervals for your prevalence values.

This was done (see table 1)

Figure 2, are the images of the possible reservoirs examples of local breeds of sheep, goats and dogs. gambiense is spelled incorrectly in the legend

Thnak you, we edited the legend of figure 2 (see fig 1 in new manuscript version)

Line 137, what does ref mean, is this figure based on molecular analyses only. The wings are are probably stars. Figure 3, spelling of Equatorial Guinea and Cameroon

Thank you, now we edited the two names. also, we removed the word ''Ref'' because it is an omission in the correction of the previous version which mean put in reference here (see line 139 in new manuscript version).

Line 192 places not place

Thank you, this was done (see line 194 in new manuscript version)

Line 202 prevalence is its own plural

Thank you, we edited (see line 204)

Line 219 check prevalence for goats and small ruminants, according to the table it should be goats 16% and small ruminants 18%. Note you have no significant difference between small ruminants and dogs so they are essentially the same. One thing you might want to note is the length of hair on the various animals as if sheep are quite hairy then this might make it more difficult for tsetse to take a meal

We agree with the reviewer that the prevalence is 16% for goats and small ruminants (goats and sheep) it is 18.34%. However in this section we compare the overall prevalence of dogs which is 23% to small ruminants which is 18.34%. And we say that it seems to be more infected than small ruminants but this is not significant. However, we have taken into account the reviewers' suggestions and comments in the new version (see line 214-216 and line 222-224).

Line 220 there is mention of cities, how urban/rural are the study locations

In Gabon, certain urban areas contrast with rural areas such as villages, this is the case of the cities Cocobeach, Ntoum which are considered rural environments and Port-Gentil is an urban area.

Reviewer 2 Report (New Reviewer)

Interesting information related to role of domestic animals in the life cycle and transmission of Trypanosoma spp. in the central Africa/Gabon.

I am of an opinion that the article fits into scope of Pathogens and could be published after major corrections.

1. Please standardize the accuracy of parasitological indicators - one time is to the first, another time to the second decimal place, e.g. lines 29-31: „that 19.36%...”  and „…21%...” – by the way it should be 19.34%.

2. According to  “Molinari and Moreno (2018): Trypanosoma brucei Plimmer & Bradford, 1899 is a synonym of T. evansi (Steel, 1885) according to current knowledge and by application of nomenclature rules. Systematic Parasitology)” Trypanosoma brucei is a synonym of T. evansi ?!. I think it is worth referring to this work.

3. Scientific host/parasite names (with authors and date) are obligatory; especially since the authors of the manuscript give common names like "cats" (line 132) - is it about “Felis silvestris Schreber, 1777”?. This is important because the parasites show host specificity.

4. Lines 33-34” „… T. simiae Tsavo” -  please use records consistent with the  biological nomenclature rules / International Code of Zoological Nomenclature.

5. line 43: “ …fly of the genus Glossina …“ – please add “…. Glossinia (tsetse fly).

6. lines 53, etc., etc.: “… continent in the mid-1960s (PMID: 34986176; PMID: 28750007)” – please PMID… treat as citation and include them in Bibliographic list as well.

7. Tables 2, 3, 4: tables are not clear; the values in Tables 2 and 3 are different, e.g. table 2 (last line and 6 column) and table 3 (last line and 3 column) are 19(16) and 22(18.6)-“ ??; please standardize the entries - once there is, e.g. table 2, line 3, column 4: 3 (21) that is infected (prevalence) and in the last column – 20 (9/45) that is prevalence (infected/examined) ???;  please change title in the last column – better “Total prevalence …”; city names should be in normal font; in the table 3 (last column/last line) please complete prevalence.

Instead of "N" it would be better "examined" and instead of "n" it would be better "infected".

Please correct title – better “Prevalence … by localities and …”. In addition, the titles are imprecise - it must be added that it is about the studied region - Africa / Gabon.

8. Figure 2: what does „A, B, C” mean ?

9. Line 138: “.. wings…: - ???

10. 152-154: “All species identified in our study were previously found in domestic animals [32-35]. Thus, our results corroborate those of previous studies conducted in other regions of Africa ([31, 35]” – repetition – please change/simplify to one sentence.

11. Line 213: “… in goats (17%) - ??? 16% or 18 %, please check and correct.

12. Lines 227-228: “…infection per species is low, T. simiae, T. b. brucei and T. congolense had the highest infection rate among domestic animals with 5%, 3% and 3%, …” – Table 2 shows that for other Trypanosoma species / subspecies, the infection was also 3%.

13. lines 235-236: “…risk to the health of human populations, as they can infect humans … “ – repetition, please correct.

14. Line 316: in my opinion, quoting Table S1 here is unnecessary.

15. Line 317: please correct number of tables – first 1 then 2, 3, etc..

16. Table 1:  “Parasites species” – should be “Parasite species/subspecies” or “Parasites”.

17. Lines 356-511: please correct references list according to Pathogens.

Author Response

Reviewe 2

Comments and Suggestions for Authors

Interesting information related to role of domestic animals in the life cycle and transmission of Trypanosoma spp. in central Africa/Gabon. I am of an opinion that the article fits into scope of Pathogens and could be published after major corrections.

  1. Please standardize the accuracy of parasitological indicators - one time is to the first, another time to the second decimal place, e.g. lines 29-31: „that 19.36%...” and „…21%...” – by the way it should be 19.34%.

Thank you, we edited and standardized the different value of parasitological indicators (see 29-32)

  1. According to “Molinari and Moreno (2018): Trypanosoma brucei Plimmer & Bradford, 1899 is a synonym of T. evansi (Steel, 1885) according to current knowledge and by application of nomenclature rules. Systematic Parasitology)” Trypanosoma brucei is a synonym of T. evansi ?!. I think it is worth referring to this work.

Indeed, its authors report that Trypanosoma brucei is a synonym of T. evansi. However, several studies state that T. evansi and T. equiperdum are subspecies of T. brucei (Lai et al., 2008; Carnes et al., 2015; Moreno et al., 2015; Wen et al., 2016) as are T. b. brucei, T. b. gambiense and T. b. rhodesiense. Thus, it seems prudent to keep the names of the different species and subspecies as mentioned in  our  study,

  1. Scientific host/parasite names (with authors and date) are obligatory; especially since the authors of the manuscript give common names like "cats" (line 132) - is it about “Felis silvestris Schreber, 1777”?. This is important because the parasites show host specificity.

We agree, that it is important to give the scientific names of different hosts mentioned in the manuscript. However, we are unable to provide them because the different articles mentioning the example of cats do not say which species they are ( Example https://doi.org/10.1080/00034983.1973.11686894 ; doi: 10.1186/s13071-017-2411-2).  

On the other hand, all the articles that allowed the realization of this map are quoted in the legend of the figure (see line 136).

  1. Lines 33-34” „… T. simiae Tsavo” - please use records consistent with the biological nomenclature rules / International Code of Zoological Nomenclature.

Thank you comment, now we corrected the writing of T. simiae Tsavo in all the manuscript (see lines 34, 107, 121, 155, 166 and in table 2 and table 3).

  1. line 43: “ …fly of the genus Glossina …“ – please add “…. Glossinia (tsetse fly).

Now, we added (see line 43)

  1. lines 53, etc., etc.: “… continent in the mid-1960s (PMID: 34986176; PMID: 28750007)” – please PMID… treat as citation and include them in Bibliographic list as well.

Thank you, this was done (see line 52-62)

  1. Tables 2, 3, 4: tables are not clear; the values in Tables 2 and 3 are different, e.g. table 2 (last line and 6 column) and table 3 (last line and 3 column) are 19(16) and 22(18.6)-“ ??;

We agree with the reviewer concerning its observation in Tables 2 and 3. In this new manuscript version, these tables become Table 1 and 2.  And the results observed in table 1 cannot be the same as table 2 because table 1 is the Prevalence of trypanosome infection by studies localities in Gabon and per host studied i.e. it presents the number of infections per site and host. On the other hand table, 2 is Diversity and prevalence of each trypanosome species identified in animals studied and taking into account the co-infection which means that an individual carrying two species of parasites will be taken into account twice hence the different values between table 1 and 2.

please standardize the entries - once there is, e.g. table 2, line 3, column 4: 3 (21) that is infected (prevalence) and in the last column – 20 (9/45) that is prevalence (infected/examined) ???; 

Thank you, we done (see table 1)

please change title in the last column – better “Total prevalence …”; city names should be in normal font; in the table 3 (last column/last line) please complete prevalence.

Okay, we edited the table 1 and 2. The cities were put in normal font, we changed total by total prevalence

Instead of "N" it would be better "examined" and instead of "n" it would be better "infected".

Okay, we done and considered reviewer suggestion (see the table 1)

Please correct title – better “Prevalence … by localities and …”. In addition, the titles are imprecise - it must be added that it is about the studied region - Africa / Gabon.

Okay, now we edited the tittle of tables 1 and 2 (see line 98 and line 112)

  1. Figure 2: what does „A, B, C” mean ?

Okay, we edited the figure 2 (see fig 2 in new manuscript version)

  1. Line 138: “.. wings…: - ???

We removed this word

  1. 152-154: “All species identified in our study were previously found in domestic animals [32-35]. Thus, our results corroborate those of previous studies conducted in other regions of Africa ([31, 35]” – repetition – please change/simplify to one sentence.

We agree and have deleted one of the sentences (see line 151-152).

  1. Line 213: “… in goats (17%) - ??? 16% or 18 %, please check and correct.

Okay, we corrected

  1. Lines 227-228: “…infection per species is low, T. simiae, T. b. brucei and T. congolense had the highest infection rate among domestic animals with 5%, 3% and 3%, …” – Table 2 shows that for other Trypanosoma species / subspecies, the infection was also 3%.

Okay, The values were the same because we rounded the prevalence values to the decimal point. Now, we have put back the values without rounding to show the slight difference .Thus the sentence become "infection per species is low, T. simiae, T. b. brucei and T. congolense had the highest infection rate among domestic animals with 4.74%, 3.28% and 3.28%) (see table 2),

  1. lines 235-236: “…risk to the health of human populations, as they can infect humans … “ – repetition, please correct.

Thank you, This was done (see line 234-235)

  1. Line 316: in my opinion, quoting Table S1 here is unnecessary.

Okay, we removed Table S1 in new manucript (see line 311)

  1. Line 317: please correct number of tables – first 1 then 2, 3, etc..

Thank you, we edited the number of table in new manuscript version (line 98 ; 112 ; 120 ; 312)

  1. Table 1: “Parasites species” – should be “Parasite species/subspecies” or “Parasites”.

Okay, we edited subspecies

  1. Lines 356-511: please correct references list according to Pathogens.

We corrected the references list

Round 2

Reviewer 2 Report (New Reviewer)

A few more small comments:

1. Line 127: “Figure 1. Spatial distribution of trypanosome species in different host species (A). Distribution of trypanosomes identified …” – should be „Figure 1. Spatial distribution of trypanosome species in different host species. (A) Distribution of trypanosomes identified …””.

2. Reference list:

Please correct references list according to Pathogens !

Journal Names should be abbreviated.

All scientific names should be italicized, e.g. line 366 (T. b. gambiense), line 412 (Trypanosma), line 414 (Trypanosoma evansi), line 419, etc.

 11. Courtin, F.; Kaba, D.; Rayaisse, J.B.; Solano, P. The cost of tsetse control using 'Tiny Targets' in the sleeping sickness endemic forest area of Bonon in Côte d'Ivoire: Implications for comparing costs across different settings. 2022, 16, e0010033, 383 doi:10.1371/journal.pntd.0010033

- Steve J. Torr and Alexandra P. M. Shaw are also authors of this article; article was published in Plos Neglected Tropical Diseases. Please supplemented and check out other articles.

8. Liana, Y.A.; Shaban, N.; Mlay, G.; Phibert, A. African Trypanosomiasis Dynamics: Modelling the Effects of Treatment, 375 Education, and Vector Trapping. International Journal of Mathematics and Mathematical Sciences 2020, 2020. – twice „2020” ?.

Besides, should be „African trypanosomiasis dynamics: Modelling the effects of treatment, education, and vector trapping”. – small letters. Please check also other articles.

Author Response

Comments and Suggestions for Authors

A few more small comments:

  1. Line 127: “Figure 1. Spatial distribution of trypanosome species in different host species (A). Distribution of trypanosomes identified …” – should be „Figure 1. Spatial distribution of trypanosome species in different host species. (A) Distribution of trypanosomes identified …””.

Thank you, now we corrected

  1. Reference list:

Please correct references list according to Pathogens !

Journal Names should be abbreviated.

Okay, we edited the name of all

All scientific names should be italicized, e.g. line 366 (T. b. gambiense), line 412 (Trypanosma), line 414 (Trypanosoma evansi), line 419, etc.

Now, we put all scientific names in italized

  1. Courtin, F.; Kaba, D.; Rayaisse, J.B.; Solano, P. The cost of tsetse control using 'Tiny Targets' in the sleeping sickness endemic forest area of Bonon in Côte d'Ivoire: Implications for comparing costs across different settings. 2022, 16, e0010033, 383 doi:10.1371/journal.pntd.0010033

- Steve J. Torr and Alexandra P. M. Shaw are also authors of this article; article was published in Plos Neglected Tropical Diseases. Please supplemented and check out other articles.

We agree and we completed authors list

  1. Liana, Y.A.; Shaban, N.; Mlay, G.; Phibert, A. African Trypanosomiasis Dynamics: Modelling the Effects of Treatment, 375 Education, and Vector Trapping. International Journal of Mathematics and Mathematical Sciences 2020, 2020. – twice „2020” ?.

The second 2020 corresponds to the volume accordant to the presentation of the journals belonging to the Hindawi group. However, we have corrected the presentation of this reference.

Besides, should be „African trypanosomiasis dynamics: Modelling the effects of treatment, education, and vector trapping”. – small letters. Please check also other articles.

We agree and we edited

This manuscript is a resubmission of an earlier submission. The following is a list of the peer review reports and author responses from that submission.

Round 1

Reviewer 1 Report

The paper by Boundenga et al focuses on trypanosomes detected in three animal species (sheep, goats and dogs) in Gabon

Abstract- needs to be written better, so for example

Line 21- humans are animals, so might be best to rephrase as "Thus, domestic and wild animals"

Line 24- the number of human cases could be elaborated on within the introduction.

Line 26- I do not think diagnosting is a word

Line 28 infection not Infection

Line 28 "across. 23.21%" seems to be missing some words

Line 29 show not shown (although this whole sentence could be improved)

Line 31, "seven trypanosomes" is a bit vague, it cannot be species (see comment below for the introduction

Line35- for HAT, no of HAT

Introduction

Line 40 and throughout trypanosomiasis (not a proper noun so does not need a capital, human is only capitialised as it starts sentence)

Line 41 and throughout, species names should be in italics

Line 42 humans

Line 45, is the 300,000 an estimate of how many are thought to be infected, or the confirmed cases

Line 47, there are three subspecies of T. brucei- T. brucei brucei, T. b. gambiense and T. b. rhodesiense, of which two infect humans

Line 48 populations

Line 49, given the sentence before, I do not think this sentence needs to start with "Despite this" (the sentence could be removed)

Line 58, "for the acute clinical forms of HAT in humans" could be written better

Line 55 theileri not theilerie (check others)

Line 60 rephrase the sentence, don't use "these latter"

Line 63, provide more detail on the history of sleeping sickness in Gabon, a previous paper https://www.ncbi.nlm.nih.gov/pmc/articles/PMC7543151/ does provide some detail in terms of number of cases each year. So it mentions 16 cases in 2018. You could add in some more recent numbers, also discuss the historical foci.

Results

You could include 95% confidence intervals with your prevalence data

My main concern around the paper is the identification of T. b. gambiense just through the use of sequencing. It is exacerbated with the placement of the T. b. rhodesiense sequence just below the T. vivax cluster in Figure 1. Given T. b. rhodesiense, T. b. brucei and T. b. gambiense are all Trypanozoon, I cannot see any reason why the T. b. rhodesiense isolate would appear where it does in Figure 1. Therefore suggest that the sequences are checked and the  tree is reconstructed.

Inclusion of the accession numbers of the products you have sequenced would be good to include in the figure (you mention accession numbers on line 246, but it would be good if these were added into Figure 1)

Also to confirm the presence of T. b. gambiense, given the TgsGP gene is diagnostic of T. b. gambiense I would suggest that the primers here (https://pubmed.ncbi.nlm.nih.gov/12408669/) are used to confirm T. b. gambiense, rather than through sequencing of the ITS region (you might want to also sequence products of the TgsGP PCR) .

Looking at for example this paper (https://www.ncbi.nlm.nih.gov/pmc/articles/PMC8224965/) they have sequenced trypanosomes found in humans in Chad, one of the ones found in humans was most similar to a sample from hyaena from Tanzania. Which given its location is unlikely to be T. b. gambiense. Therefore I would be wary of designating T. b. gambiense through sequencing of ITS.

line 76, given stats analysis shows no difference, this sentence could be changed to reflect this (focus on removing "more infected")

Line 83, see previous comments around species/subspecies

Line 91, this could be written better "However, dogs were more infected than sheep with 8.9%"

Lines 97-98, report prevalence of the mixed infections, rather than the percentage of overall trypanosome infections that are mixed (the denominator should bbe the number of animals sampled.

Line 100- an association could be construed as something that is significant, is  this the case

Table 2, a comma creeps in for Moyen-Ogooue (8.695 round to 8.70, check others)

Table 4 Change Percentage to Prevalence, also Host species not speceis

Figure 2, percentages could be added to the map rather than just the numerator and denominator

Discussion

Generally discussion ok, although, may need to be rewritten slightly if TgsGP primer sets are used

Line 175, I am unsure what you mean here by pets, is this referring to the dogs only. Would dogs be considered pets within the area.

Line 176, where has 18.61 (51/228) come from

Line 184, this could be written better "Also, dogs (23.21%) were high infection rate that to small"

Line 195- for surveillance (no of)

Line 211, why are mixed infections dangerous

I am unsure if the methods should be sandwiched between the discussion and conclusion, possibly place after the conclusion

Methods

From reading the methods, two primer sets were used, were the results in agreement with the two different sets (unclear if both primer sets were used on all samples or one with some and the second on the rest), or were they radically different.

Reviewer 2 Report

Please see attached word document

Reviewer 3 Report

This paper cannot be published in its current form because it is so packed with errors, misjudgements and lack of rigor that it is frustrating for this reviewer to just read it. I don't believe Prugnolle ever read this paper. I have found dozens of mistakes - just abstract: Infection (why capital letter?), varied across (across what?), our results shown..., Trypanosma - miss-spelled repeatedly; are the authors dysgraphic or they did not bother to read it?

Introduction starts with the same sentence as the abstract. Once it is Trypanosoma genus, elsewhere genus Glossina (flipped order, no Italics etc.). And so it goes in pretty much every sentence. The same information is repeated in Introduction and Discussion, use French abbreviation (18S ARNs is probably 18S rRNA, why not use Russian cyrilics?), some references have full names some abbreviations. Even more importantly - it is totally unclear how they distinguish between T. b. brucei and T. b. gambiense, they provide absolutely no information in the Materials and Methods regarding this principal issue etc.